# Interplay between *de novo* and salvage pathways of GDP-fucose synthesis

**Edyta Skurska, Mariusz Olczak** *

Department of Biochemistry, Faculty of Biotechnology, University of Wroclaw, Wroclaw, Poland

* mariusz.olczak@uwr.edu.pl

## Abstract

GDP-fucose is synthesised via two pathways: *de novo* and salvage. The first uses GDP-mannose as a substrate, and the second uses free fucose. To date, these pathways have been considered to work separately and not to have an influence on each other. We report the mutual response of the *de novo* and salvage pathways to the lack of enzymes from a particular route of GDP-fucose synthesis. We detected different efficiencies of GDP-fucose and fucosylated structure synthesis after a single inactivation of enzymes of the *de novo* pathway. Our study demonstrated the unequal influence of the salvage enzymes on the production of GDP-fucose by enzymes of the *de novo* biosynthesis pathway. Simultaneously, we detected an elevated level of one of the enzymes of the *de novo* pathway in the cell line lacking the enzyme of the salvage biosynthesis pathway. Additionally, we identified dissimilarities in fucose uptake between cells lacking TSTA3 and GMDS proteins.

**Data Availability Statement:** All relevant data are within the manuscript and its Supporting Information files.

**Funding:** 1. National Science Centre (Narodowe Centrum Nauki, NCN), Poland, grant number 2022/45/N/NZ3/00093 (to ES) and 2. National Science

## Introduction

Fucose, a crucial component of many glycoconjugate structures, plays a pivotal role in numerous biological processes in mammalian cells. These fucosylated structures are involved in crucial functions such as cell adhesion, tissue development, angiogenesis, fertilisation, malignancy, and tumour metastasis [1–3]. Increasingly, up/downregulation of fucosylation is found in cancer cells [4–6]. Some of them can cause tumour multidrug resistance (MDR) [7, 8]. Besides, it was shown that core fucosylation induced epithelial-mesenchymal transition (EMT) in lung cancer cells, promoting cell migration [9]. Fucosylation also takes part in the development of immune cells [10]. Abnormalities in terminal fucosylation were pointed out as a hallmark of inflammatory macrophages in rheumatoid arthritis [11].

Only its active form, GDP-fucose, is used to synthesise oligosaccharides. In mammals, GDP-fucose is produced via two separately working biosynthesis mechanisms, *de novo* and salvage pathways [1]. The *de novo* pathway converts GDP-mannose to GDP-fucose in a three-step enzymatic reaction. Firstly, GDP-mannose 4,6-dehydratase (GMDS) converts GDP-mannose to GDP-4-keto-6-deoxymannose [12], and then GDP-keto-6-deoxymannose-3,5-epimerase (TSTA3), an enzyme of dual activity of epimerase/reductase, transforms it to GDP-fucose [13]. In the salvage pathway, L-β-fucose is phosphorylated by fucokinase (FCSK) [14], and fucose-1-phosphate is converted by GDP-fucose pyrophosphorylase (FPGT) to GDP-fucose

Centre (Narodowe Centrum Nauki, NCN), Poland, grant number 2023/51/B/NZ3/00810 (to MO) (the second, new source of financing) We declare that the funders had no role in study design, data collection and analysis, decision to publish, or preparation of the manuscript in both grants.

**Competing interests:** The authors have declared that no competing interests exist.

**A** *De novo*

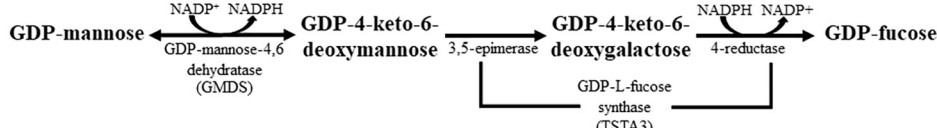

**B** Salvage

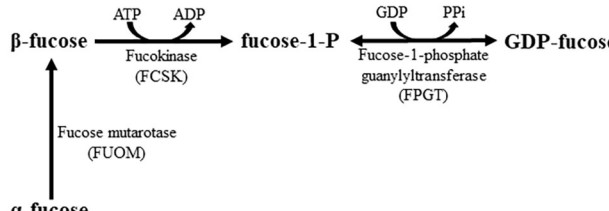

**Fig 1. Graphical presentation of GDP-fucose biosynthesis pathways in mammalian cells.** (A) scheme presenting the *de novo* biosynthesis pathway of GDP-fucose. (B) diagram illustrating the fucose-derived pathway of GDP-fucose synthesis.

(Fig 1) [15]. Most GDP-fucose is produced via the *de novo* pathway, while free L-fucose is reused for the salvage pathway [16, 17].

Free L-fucose is a substrate utilised by fucokinase. It could originate from an extracellular space or lysosomal degradation of fucosylated glycans [14]. Alpha-L-fucosidase (FUCA1) hydrolyses the alpha-linked fucose joined to the N-acetylglucosamine or galactose moieties of glycoproteins in lysosomes [18, 19]. The reaction liberates the α-anomer of L-fucose, which cannot be utilised by fucokinase, as this enzyme uses only the β-version of L-fucose [20]. Fucose mutarotase converts fucose from the α to the β anomer [21]. L-fucose, in its β-anomer, can be obtained from outside the cell. There are at least two mechanisms of uptake of L-fucose by mammalian cells [22]; one is based on macropinocytosis, which is initiated by the sensing of extracellular calcium by the G protein-coupled receptor, calcium-sensing receptor, CaSR [23]. Recently, the second mechanism of fucose uptake, depending on glucose transporter 1 (GLUT1) activity, was described [22].

To illustrate the importance of the fucosylation of antibodies, the depletion of genes encoding enzymes of the *de novo* GDP-fucose biosynthesis pathway was applied to produce afucosylated antibodies. The lack of fucose in antibodies is well known to enhance the antibody-dependent cellular cytotoxicity [24–28]. These studies have shown a complete loss of fucosylated N-glycans decorating synthesised antibodies and indicated that fucosylation of antibodies produced by cell lines deficient in TSTA3 or GMDS can be regulated equally by fucose supplementation in various concentrations. Moreover, there is no information on whether the lack of enzymes taking part in the mannose-derived GDP-fucose biosynthesis pathway affects salvage biosynthesis proteins.

In this study, we employed the CRISPR/Cas9 system to select cell lines deficient in TSTA3 (TSTA3KO), GMDS (GMDSKO) and FCSK (FCSKKO) in the human embryonic kidney 293T (HEK293T) cell line. We found that cells lacking the TSTA3 enzyme produced enormously high amounts of GDP-fucose upon fucose supplementation, whereas it does not happen in cells deficient in GMDS protein. We revealed the mutual regulation of the work by *de novo* and salvage pathway enzymes. We observed that the level of fucokinase was elevated in

cells lacking TSTA3 protein upon fucose supplementation but not in cells lacking GMDS protein. Moreover, we found that the protein level of TSTA3 was raised in cells lacking FCSK enzyme. We explored differences in fucose uptake between studied cell lines and the potential mechanism leading to that.

## Materials and methods

### Cell culture maintenance and gene inactivation

Cells of the HEK293T cell line purchased from ATTC (American Type Culture Collection) were cultured in Dulbecco's Minimum Eagle Medium (DMEM High Glucose, Biowest) supplemented with 10% fetal bovine serum, 100 U/ml penicillin and 100 μg/ml streptomycin. Cell cultures were kept at 5% $CO_2$ and 37˚C.

Inactivation of the FCSK, GMDS and TSTA3 genes was performed using a ready-to-use CRISPR/Cas9 system provided by Santa Cruz Biotechnology. According to the manufacturer's protocol, a mixture of human TSTA3 double nickase plasmids (sc-408777-NIC), human GMDS double nickase plasmids (sc-410794-NIC) or human FCSK double nickase plasmids (sc-409449-NIC) was transfected into the HEK293T wild-type cell line. For transfection, the transfection reagent FUGENE HD was applied, according to the manufacturer's instructions. Then, cells were cultured in complete media supplemented with 1 μg/ml puromycin for around three weeks. After that, clones were isolated. Total RNA was extracted from isolated clones and subjected to RT-PCR analysis using gene-specific primers. As a control, total RNA isolated from wild-type cells was used. Analysis of genomic DNA isolated from cells deficient in TSTA3, FCSK, or GMDS proteins was also performed using gene-specific primers (S1 Table in S2 File). The last step of verification of gene inactivation was western blotting.

### Generation of cell lines overexpressing FCSK and FPGT proteins

The cDNAs encoding human fucokinase (NCBI accession number NM_145059.3) and fucose-1-phosphate guanylyltransferase (NCBI accession number NM_003838.5) were amplified from total HEK293T wild-type RNA. Forward primers contained gene sequences encoding HA protein for fucokinase or c-myc protein for FPGT enzyme. Used primers are listed in S2 Table in S2 File. Amplified genes were cloned into the pSelect-zeo-mcs plasmid (Invivogen) using the restriction cloning technique. The restriction enzymes SalI and NheI were applied in the case of FCSK restriction cloning, and the restriction enzymes SalI and AfeI were used for FPGT restriction cloning.

According to the manufacturer's protocol, HEK293T GMDSKO and TSTA3KO cell lines were transfected with obtained vectors encoding fucokinase and FPGT using the FUGENE HD transfection reagent. Then, cells were cultured in complete media containing additional 400 μg/mL zeocin for about one month. The transfection efficiency was ~95% for all overexpression; therefore, we decided not to isolate clones but to work with mixtures after transfection and selection using antibiotics.

### Western blotting

Harvested cells were lysed, and the cell lysates were separated by SDS-PAGE electrophoresis. They were then transferred onto a nitrocellulose membrane (Whatman), blocked, incubated with primary and secondary antibodies, and detected as described previously [29]. Alternatively, chemiluminescence signals were detected using the ChemiDoc imaging system (Bio-Rad Laboratories). The antibodies used in Western blotting analyses are listed in S3 Table in S2 File.

### Analysis of α-1,6 fucosylated N-glycans

The method was described previously [30]. Briefly, cells, unsupplemented and supplemented with fucose, were harvested and lysed. Proteins from cell lysates were concentrated by acetone precipitation and resolved in a denaturing buffer for the enzymatic removal of N-glycans. Then, after purification on a graphite solid phase extraction (SPE) column, they were labelled with 2-AB. Purified, 2-AB labelled and dried N-glycan pools were treated with neuraminidase A, β-1-4 galactosidase S and β-N-acetyl-glucosaminidase S (all enzymes were purchased from New England Biolabs). After digestion, N-glycans were separated by RP-HPLC, and the percentage of fucosylated structures was calculated as described previously [30].

### Analysis of GDP-fucose concentration in cells

Cells, unsupplemented and supplemented with fucose, were harvested, counted using a Trypan Blue (Thermo Fisher Scientific) reagent and frozen. Frozen cell pellets were subjected to nucleotide sugar extraction using a previously published method [30]. As reported before, purified nucleotide sugars were separated in ion-pairing, reverse-phase HPLC [30].

### L-fucose supplementation

In most fucose supplementation experiments, cell lines were cultured in complete media with the addition of 5 mM L-fucose for 24 h. In the study where fucose uptake was investigated, L-fucose concentrations of 0.1 mM, 1 mM and 5 mM were applied. L-fucose concentrations of 10 μM, 50 μM, 0.1 mM, 1 mM and 5 mM were used in the experiment analysing the influence of stimulation of the salvage biosynthesis pathway in TSTA3KO and GMDSKO upon feeding in different fucose amounts. Conditions for fucose supplementation were adopted from our previous study [30].

### Radioactive labelling of N-glycans and nucleotide sugars

Cells were cultured in a complete medium with the addition of 4 μCi/ml L-[6-³H] fucose (American Radiolabeled Chemicals, specific activity 60 Ci/mmol), for analysis of N-glycans and 4 μCi/ml L-[5-6-³H] fucose (American Radiolabeled Chemicals, specific activity 60 Ci/mmol) for 24 hours, to analyse the GDP-fucose cellular synthesis. Then, cells were collected and subjected to N-glycan isolation or nucleotide sugar extraction, as described above.

### Fucose uptake assay

The assay was performed as previously reported [30]. Briefly, cells after supplementation with fucose in 0.01 mM, 1 mM and 5 mM concentrations in cell culture media for 24 h were scraped, washed twice with phosphate buffer saline, and counted. Cell pellets were resuspended in Milli-Q water and sonicated. The excess proteins were removed with perchloric acid. Fucose concentration was analysed in the supernatant, as described previously [30]. A standard curve was prepared for calculations of fucose concentration in cells.

### Statistical analysis

Statistical analyses were performed using GraphPad Prism 8. The percentage of fucosylation structures and concentration of GDP-fucose were analysed in three biological replicates, except the concentration of GDP-fucose upon supplementation with different amounts of fucose. One-way ANOVA was used to analyse data with the Tukey post-hoc test. Statistical parameters, including data plotted (mean ± SEM) and P values, are presented in detail in the Fig legends. Western blots were run at least in two biological replicates. One-way ANOVA was

used to analyse data with the Tukey post-hoc tests or Welch's t-test. Statistical parameters, including data plotted (mean ± SEM) and P values, are presented in detail in the Fig legends.

## Results

### Generation of TSTA3, GMDS and FCSK knockouts

To develop cell models for our study, single knockouts of TSTA3 (TSTA3KO), GMDS (GMDSKO) and FCSK (FCSKKO) were established in the human cell line HEK293T. We employed the CRISPR/Cas9 double nickase system to obtain cell lines deficient in investigated proteins. The absence of an inactivated enzyme in generated cells was confirmed by western blotting using anti-TSTA3, anti-GMDS and anti-FCSK antibodies (S1A–S1C Fig). Additionally, the knockouts were confirmed by RT-PCR performed on total RNA and PCR performed on genomic DNA (S1D–S1E Fig). In further experiments, two clones out of each gene inactivation were used.

### Quantification of intracellular GDP-fucose and analysis of the level of fucosylated α-1,6 N-glycans

It is assumed that ~90% of the total pool of GDP-fucose originates from the action of the *de novo* biosynthesis pathway, and another ~10% is produced by the salvage biosynthesis pathway [16, 17]. GDP-fucose is the substrate for fucosyltransferases, which incorporate fucose into glycoconjugates, mostly N-glycans. Therefore, we quantified the intracellular GDP-fucose concentration and analysed the level of the most common fucosylation in N-glycans (α-1,6 fucose) with and without external fucose supplementation (5 mM, 24 h) in all generated knockouts.

The GDP-fucose concentration in unfed TSTA3KO cells was estimated as 0 μM. The overlapping peak with a very small GDP-fucose signal made the readout harder (S2A–S2C Fig). However, upon fucose supplementation, this concentration increased to ~2400 μM. In contrast, in wild-type cells, this level was ~400 μM (Fig 2A). From analysis of the level of fucosylated structures, we could observe a decrease in the percentage of fucosylation from ~80% in wild-type cells to ~3% in TSTA3KO cells. Upon fucose supplementation, this level increased to almost 51% (Fig 2B). The dramatic production of GDP-fucose caused a great improvement in the amount of fucosylated glycans. However, it did not influence the incorporation of fucose at similar levels to wild-type cells.

From the analysis of the cytosolic concentration of GDP-fucose in the GMDSKO cell line (GMDS is the first enzyme of the *de novo* biosynthesis pathway), we could conclude that the level of nucleotide sugar in unsupplemented cells is ~3 μM and rises to a concentration of ~500 μM, a similar level to fucose-supplemented wild-type cells (Fig 2C). Based on these findings, we concluded that in cells lacking in TSTA3 protein, GDP-fucose synthesis is more pronounced than in cells lacking in GMDS protein and wild-type cells. It is worth mentioning that both proteins are involved in the same biosynthesis pathway. It seems that such behaviour may be caused by a loss of control over the action of the salvage pathway in TSTA3KO cells but not in GMDSKO cells. The level of fucosylated structures in GMDSKO cells dropped to ~3% from ~80% in wild-type cells (Fig 2D). However, upon fucose supplementation, restoration of fucosylation was observed, even more pronounced than in TSTA3KO cells, but still not similar to the level in wild-type cells.

From the analysis of GDP-fucose concentration in the FCSKKO HEK293T cell line, we could conclude that in unsupplemented cells, the level of nucleotide sugar is approximately 12 μM, quite similar to the level in unfed wild-type cells (~16 μM). It does not increase upon

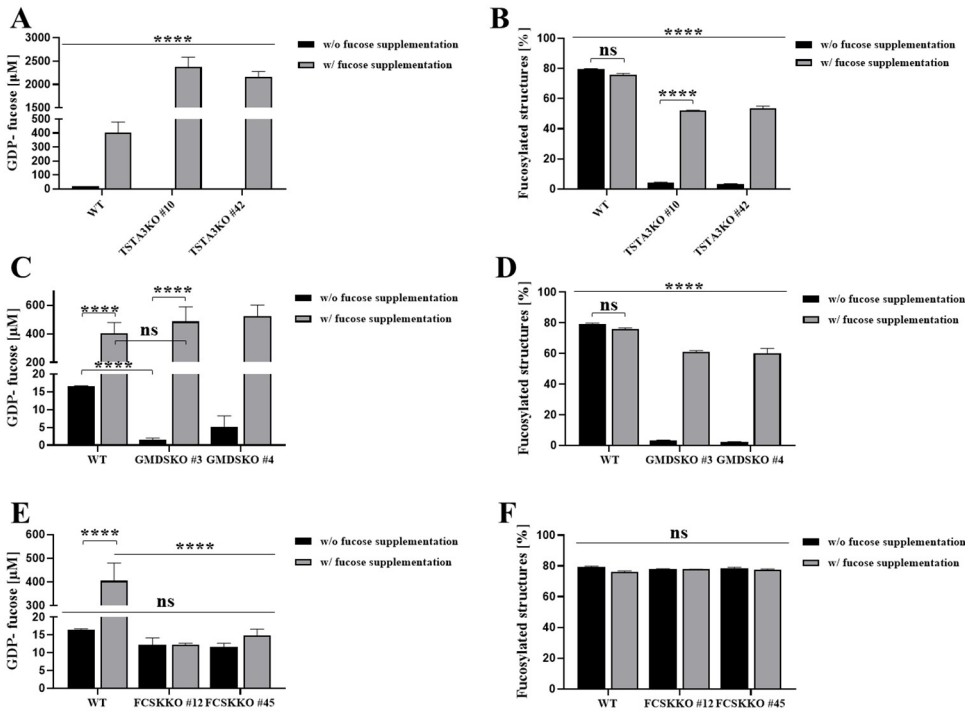

**Fig 2. Analysis of intracellular concentration of GDP-fucose and the level of fucosylated structures in the control and fucose-fed knockouts generated in the HEK293T cell line.** (A) and (B) present intracellular concentration of GDP-fucose and the level of fucosylated structures in the TSTA3KO cell line, respectively. (C) and (D) present the intracellular concentration of GDP-fucose and the level of fucosylated structures in the GMDSKO cell line, respectively. (E) and (F) show the intracellular concentration of GDP-fucose and the level of fucosylated structures in the FCSKKO cell line, respectively. Data are presented as mean ± SEM. Each sample was run in three biological replicates, ns, not significant and ****, p < 0.0001, as determined using one-way ANOVA with the Tukey post hoc test.

fucose feeding, which confirms the lack of a non-additional GDP-fucose biosynthesis pathway which uses fucose as the starting substrate (Fig 2E). Inactivation of the gene encoding fucokinase did not affect the level of α-1,6 fucosylated N-glycans (Fig 2F).

## Differential effect of various fucose concentrations on GDP-fucose synthesis in GMDSKO and TSTA3KO cell lines

Taking into account that upon 5 mM fucose supplementation, synthesis of GDP-fucose in TSTA3KO cells and GMDSKO cells differed significantly, we tested whether smaller concentrations of supplemented fucose would give a similar effect. We fed TSTA3KO and GMDSKO cells in a broad range of external fucose concentrations, between 10 μM and 5 mM. The analysis of nucleotide sugar concentration showed significant changes between analysed TSTA3KO and GMDSKO cell lines. The addition of fucose in the concentration of 50 μM already caused an increase in GDP-fucose levels in TSTA3KO cells to the rate between ~100 and 600 μM, depending on the isolated cell clone; meanwhile, barely any effect was observed in the synthesis of GDP-fucose in GMDSKO cells in the same conditions. For the GMDSKO cell line, a concentration of 1 mM of fucose could only enhance nucleotide sugar synthesis from a level of ~3 μM to ~100 μM. The same sugar concentration, 1 mM, in the TSTA3KO cell line caused the synthesis of GDP-fucose in the concentration of ~1000 μM (Fig 3A), approximately 10 times higher. Interestingly, differential synthesis of GDP-fucose in studied cell lines did not significantly affect the production of fucosylated glycoconjugates (Fig 3B). For both GMDSKO

**A**

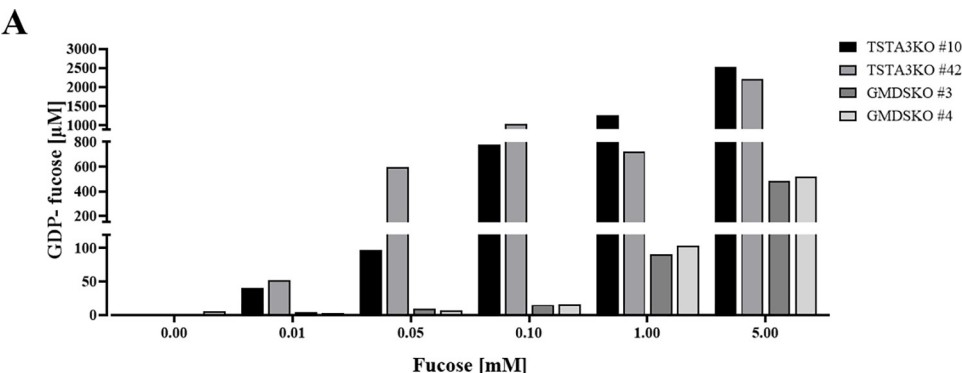

**B**

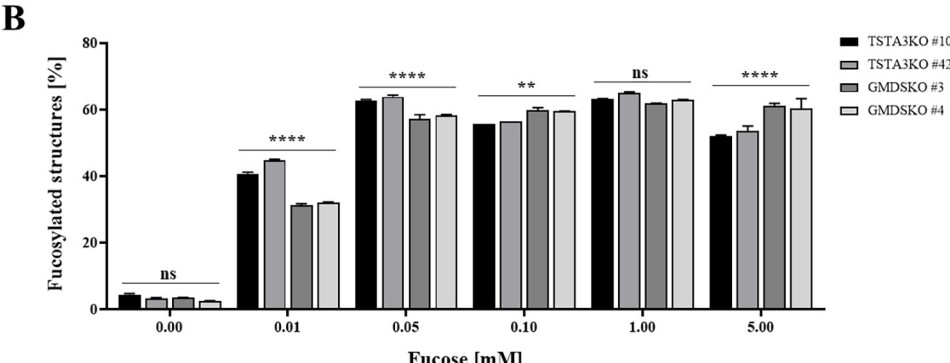

**Fig 3. Analysis of the effect of supplementation of the TSTA3KO and the GMDSKO HEK293T cell lines with different fucose concentrations.** (A) quantification of intracellular GDP-fucose concentration. (B) quantification of the percentage of the core-fucosylated N-glycan structures; each sample was run in three biological replicates. Data are represented as mean ± SEM. Each sample was run in three biological replicates, ns, not significant; **p < 0.01; and ****p < 0.0001, as determined using one-way ANOVA with the Tukey post hoc test.

and TSTA3KO cell lines, even small doses of fucose in cell culture media led to improvement in the level of fucosylated structures, e.g. supplementation at a fucose concentration of 10 μM resulted in an increased amount of fucosylated N-glycans, from ~3% to 40% and from ~3% to ~32%, for TSTA3KO and GMDSKO cell lines, respectively. One possible explanation for this phenomenon is the Km value, which is reached by the SLC35C1 protein, a GDP-fucose transporter in the Golgi apparatus. It was reported that the Km value of nucleotide sugar transporters varies from 1 μM to 10 μM of nucleotide sugar [31]. To check if the protein level of SLC35C1 in GMDSKO and TSTA3KO cell lines was unchanged compared to wild-type cells, we did western blotting using a very specific antibody against SLC35C1 transporter. No variations were observed in TSTA3KO and GMDSKO cells, supplemented and unsupplemented in fucose, relative to wild-type cells (S3A, S3B Fig).

### A deficiency of TSTA3 protein of the *de novo* biosynthesis pathway changes the level of FCSK from the salvage biosynthesis pathway

We examined the levels of FCSK and FPGT proteins, enzymes taking part in the salvage GDP-fucose biosynthesis pathway, in TSTA3KO and GMDSKO cell lines as a potential explanation of variation in the concentration of GDP-fucose in particular cell lines. It was possible that due to enhanced levels of FCSK or FPGT in TSTA3KO cells, we observed abnormal GDP-fucose production in that cell line. By employing the western blotting technique, we observed that FPGT protein level changed in TSTA3KO and GMDSKO cell lines, compared to wild-type

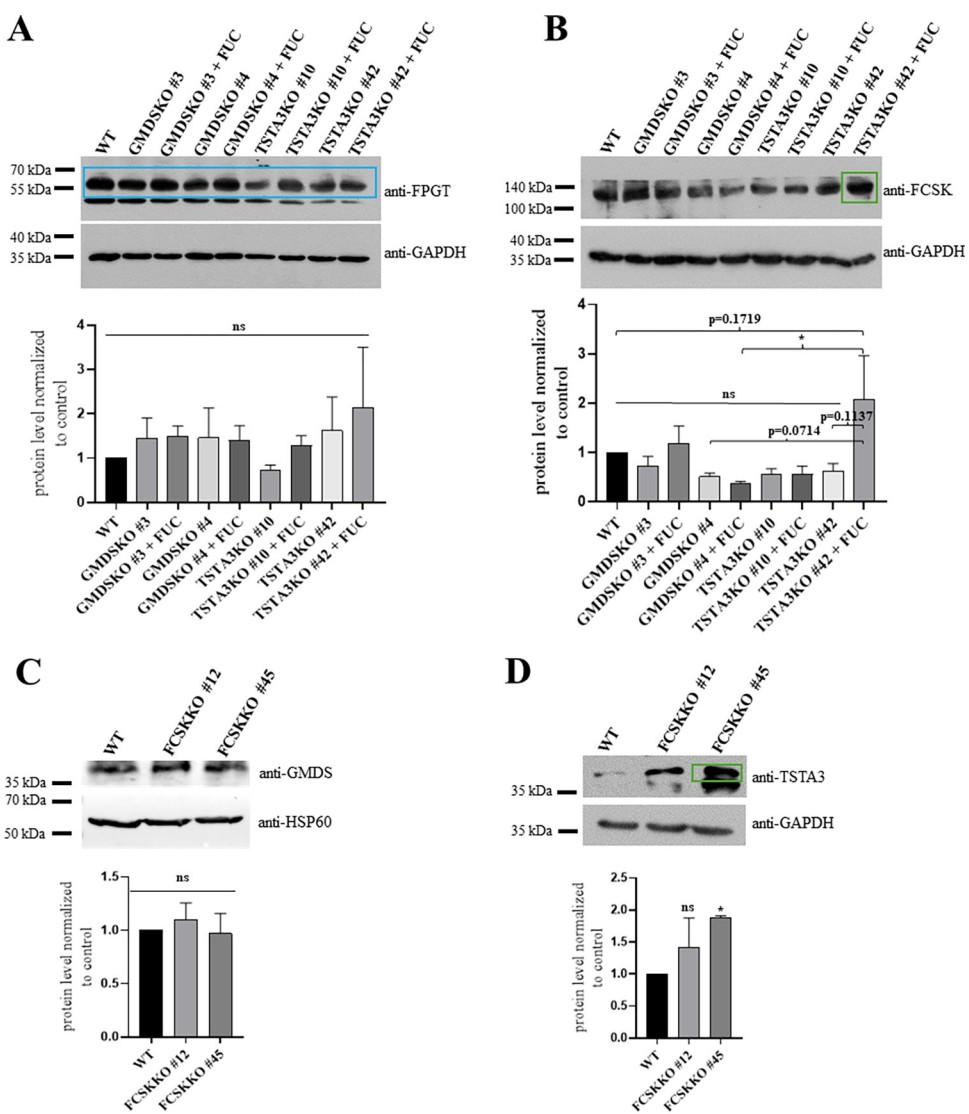

**Fig 4. Changes in levels of proteins involved in GDP-fucose synthesis.** (A) FPGT western blotting analysis of the wild-type, TSTA3KO and GMDSKO HEK293T cells without and after supplementation with fucose. As a loading control, the anti-GAPDH antibody was used. (B) FCSK western blotting analysis of the wild-type, TSTA3KO and GMDSKO HEK293T cells after supplementation with or without fucose. As a loading control, the anti-GAPDH antibody was used. The main change in FCSK protein level in TSTA3KO cell line was indicated with a green box. (C) GMDS western blotting analysis of the wild-type and FCSKKO HEK293T cells. As a loading control, the anti-GAPDH antibody was used. (D) TSTA3 western blotting analysis of the wild-type and FCSKKO HEK293T cells. As a loading control, the anti-GAPDH antibody was used. The main change in TSTA3 protein level in FCSKKO cell line was indicated with a green box. For (A) and (B), ns, not significant; *, p < 0.05 as determined using one-way ANOVA with the Tukey post-hoc test. P values that trend to be statistically significant are also shown. For (C) and (D), ns, not significant; *, p < 0.05 as determined using Welch's t-test. Data are represented as mean ± SEM. Each sample was run at least in two biological replicates.

cells, whether cells were supplemented with fucose or not (Fig 4A). However, the FCSK protein level was elevated in one of the clones of TSTA3KO cells, regardless of fucose supplementation. Moreover, that level was higher than in GMDSKO cells (Fig 4B). In agreement with that finding, we found unchanged levels of GMDS protein in cell lines deficient in FCSK enzyme compared to wild-type cells (Fig 4C) and raised levels of TSTA3 in one of the clones (Fig 4D).

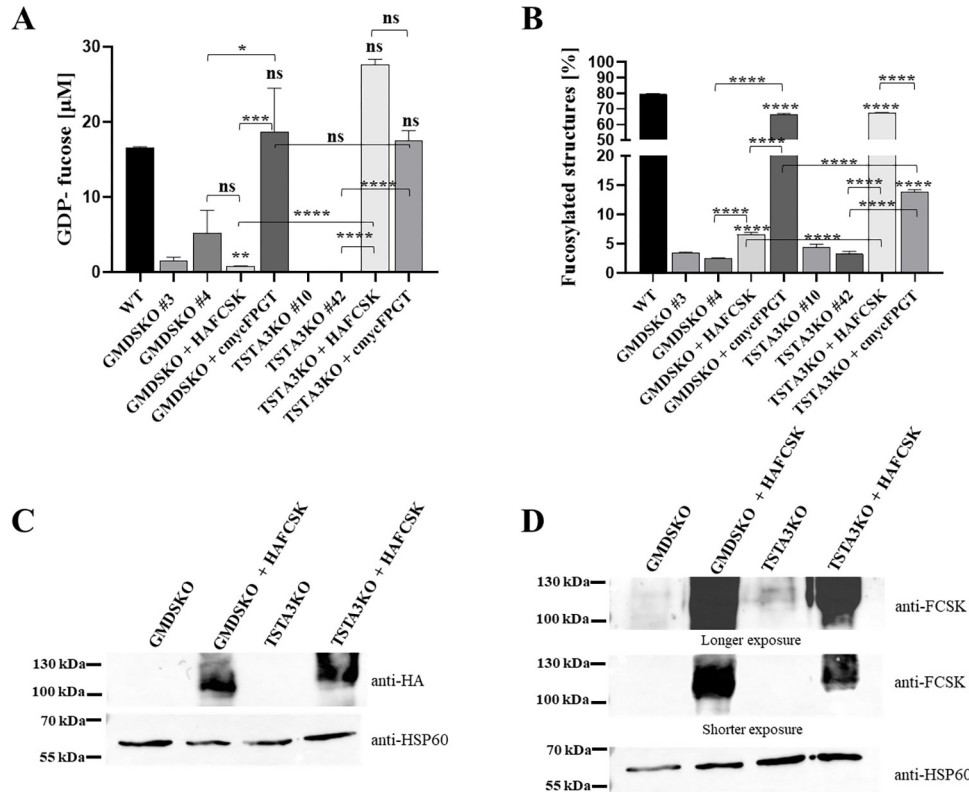

**Fig 5. Overexpression of FCSK and FPGT in TSTA3KO and GMDSKO HEK293T cell lines.** (A) intracellular concentration of GDP-fucose of wild-type, TSTA3KO and GMDSKO HEK293T cells, and in cells with overexpressed FPGT and FCSK proteins (tagged with HA and cmyc, respectively) in knockouts. (B) quantification of the percentage of fucosylated structures in wild-type, TSTA3KO and GMDSKO HEK293T cells and in cells with overexpressed FPGT or FCSK proteins in knockouts. (C) western blotting of HA in GMDSKO and TSTA3KO cells and overexpressed FPGT and FCSK in cell lines deficient in TSTA3 and GMDS enzymes. (D) western blotting of fucokinase in GMDSKO and TSTA3KO cells and overexpressed FPGT and FCSK in cell lines deficient in TSTA3 and GMDS enzymes. As a loading control, the anti-HSP60 antibody was used. For (A) and (B), ns, not significant; *, $p < 0.05$; **, $p < 0.01$; ***, $p < 0.001$; ****, $p < 0.0001$ as determined using one-way ANOVA with the Tukey post-hoc test. Data are represented as mean ± SEM. Each sample was run in three biological replicates.

Our data showed an unchanged level of FPGT protein in both cell lines and a raised level of FCSK protein in cells deficient in TSTA3 enzyme. For a more in-depth investigation of whether proteins involved in the mannose-derived GDP-fucose biosynthesis pathway could regulate the action of particular enzymes taking part in salvage GDP-fucose biosynthesis, we overexpressed FCSK or FPGT proteins (tagged with HA and cmyc, respectively) in cells lacking TSTA3 or GMDS proteins. Analysis of intracellular GDP-nucleotide sugar concentration revealed changes in the action of fucokinase. Overexpression of fucokinase greatly improved GDP-fucose concentration in TSTA3KO cells, from ~0 μM to ~ 27 μM, whereas no change was observed for GMDSKO cells (Fig 5A). Overexpression of FPGT increased the concentration of GDP-nucleotide sugar in both TSTA3KO and GMDSKO cell lines, and the difference between them was not significant. Quantification of the level of fucosylated structures showed a quite similar effect (Fig 5B). Overexpression of FCSK influenced the level of fucosylated N-glycans significantly in both examined cell lines; nevertheless, the elevation of fucokinase level promoted the incorporation of fucose into the structure of glycans in the TSTA3KO cell line more than in the GMDSKO cell line. For verification that an equal level of fucokinase was overproduced in GMDSKO and TSTA3KO cell lines, western blotting using an anti-HA

antibody (Fig 5C) was applied. Additionally, it excluded the effect of uneven participation of overexpression of tagged FCSK. Moreover, to show that overexpression significantly enhanced the level of fucokinase in cell lines deficient in TSTA3 and GMDS enzymes compared to knockouts, western blotting of endogenous fucokinase in TSTA3KO and GMDSKO cells and, additionally, in TSTA3KO and GMDSKO cell lines with overproduced fucokinase was performed (Fig 5D). It is another piece of evidence that enzymes of the *de novo* biosynthesis pathway may control GDP-fucose production via the salvage biosynthesis pathway. Moreover, that regulation did not contribute equally to both enzymes TSTA3 and GMDS. Overexpression of FPGT significantly affected the level of fucosylated N-glycans in both TSTA3KO and GMDSKO cells. However, the FCSK level elevation boosted fucose incorporation into glycan structures in the GMDSKO cell line slightly more than in the TSTA3KO cell line.

## Fucose enters more efficiently into the TSTA3KO cells than the GMDSKO cell line

The substrate for FCSK, the first enzyme of the salvage biosynthesis pathway of GDP-fucose, is free fucose, which originates from the environment or lysosomal degradation of glycans [14]. In terms of the examination of whether differential synthesis of GDP-nucleotide sugar by the TSTA3KO and the GMDSKO cells could arise from various fucose uptake from cell culture media, we tested fucose concentration in cells after fucose supplementation. We fed cells, i.e. WT, TSTA3KO, GMDSKO and FCSKKO, with three different concentrations of fucose in cell culture media, i.e. 0.1 mM, 1 mM and 5 mM, for 24 h, and then by employing colourimetric assay we measured the concentration of absorbed sugar. Differences in fucose uptake were noted at feeding points of 0.1 mM and 5 mM fucose. TSTA3KO cells absorbed similar amounts of fucose compared to wild-type cells in all tested concentrations of fucose in cell culture media (Table 1). On the other hand, GMDSKO cells took up minor fucose doses compared to wild-type and TSTA3KO cells upon supplementation with 0.1 mM and 5 mM fucose. Interestingly, for FCSKKO cells, a decrease in fucose uptake was not observed in any of the applied concentrations of this monosugar, although only fucokinase could use it as a substrate for GDP-fucose synthesis. Therefore, it is plausible that a lack of enzyme taking part in the conversion of fucose to GDP-fucose might regulate sugar uptake from outside the cell. Instead of that, it seems that the *de novo* biosynthesis pathway may play a key role in controlling fucose uptake from the environment.

We also fed wild-type, TSTA3KO and GMDSKO cells with radioactively labelled 3H-fucose to compare the efficiency of fucose absorption from outside the cells at small, close to physiological (or even lower) monosugar concentrations. The incorporation of radioactively labelled fucose into GDP-fucose in GMDSKO cells was slightly lower than in TSTA3KO cells (Fig 6A),

**Table 1. Fucose concentration (expressed in μM) in wild-type, TSTA3KO, GMDSKO and FCSKKO HEK293T cell lines.**

| Cell line | Supplemented fucose in medium | | |
|---|---|---|---|
| | 0.1 mM | 1.0 mM | 5.0 mM |
| WT | 27 | 459 | 2328 |
| TSTA3KO #10 | 42 | 392 | 2132 |
| TSTA3KO #42 | 40 | 473 | 1819 |
| GMDSKO #3 | 7 | 494 | 815 |
| GMDSKO #4 | 8 | 296 | 523 |
| FCSKKO #12 | 60 | 330 | 2177 |
| FCSKKO #45 | 35 | 305 | 2645 |

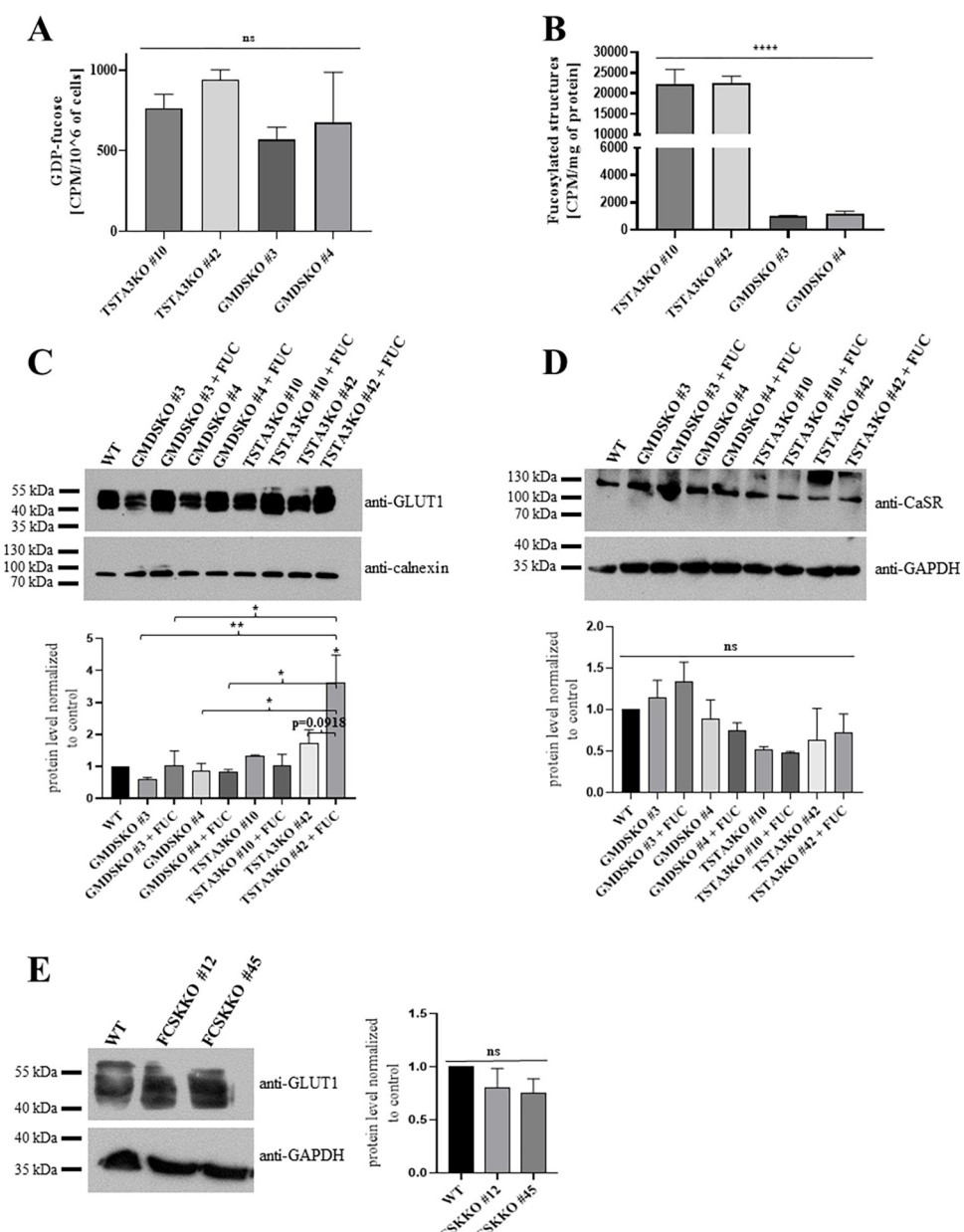

**Fig 6. Influence of TSTA3, GMDS and FCSK protein depletion on the level of proteins engaged in fucose absorption.** (A) intracellular concentration of GDP-fucose measured in TSTA3KO and GMDSKO HEK293T cell lines after supplementation with 3H-fucose. (B) result of $^3$H-fucose labelled N-glycans isolated from the TSTA3KO and the GMDSKO HEK293T cells. For (A) and (B); ns, not significant; ****, p < 0.0001 as determined using one-way ANOVA with the Tukey post-hoc test. Data are presented as mean ± SEM. Each sample was run at least in three biological replicates. Without and after supplementation with fucose, the wild-type, TSTA3KO and GMDSKO HEK293T cells were subjected to western blot analysis of (C) GLUT1, (D) CaSR. Anti-calnexin and anti-GAPDH antibodies, respectively, were used as the loading control. (E) wild-type and FCSKKO cells were employed for western blot analysis of GLUT1. For the presentation of equal loading of samples, the anti-GAPDH antibody was used. For (C) and (D), ns, not significant; *, p < 0.05; **, p <0.01 as determined using one-way ANOVA with the Tukey post-hoc test. P values that trend to be statistically significant are also shown. For (E), ns, not significant as determined using Welch's t-test. Data are represented as mean ± SEM. Each sample was run at least in two biological replicates.

but the amount of α-1,6 fucosylated N-glycan structures was enormously higher (Fig 6B). The difference between cell lines in the incorporation of radioactively labelled fucose probably results from the observed various fucose uptake by GMDSKO and TSTA3KO cell lines, or, in TSTA3KO cells, the mechanism of delivery of GDP-fucose by the route which works only under very low (nanomolar) concentrations of this nucleotide sugar works more efficiently than in GMDSKO cells [30].

Two mechanisms of absorption of fucose into mammalian cells have been proposed. The first is macropinocytosis; the second depends on the GLUT1 transporter action [22]. Since the differences in fucose imbibition were visible, we sought to determine which mechanism may be affected by the absence of TSTA3 or GMDS proteins. Western blotting of GLUT1 showed an almost unchanged level of this protein in GMDSKO and TSTA3KO cells without fucose supplementation compared to wild-type cells (Fig 6C). However, upon fucose feeding, in TSTA3KO cells, the level of GLUT1 was higher than in GMDSKO cells treated and not treated with fucose. Additionally, we checked the glucose transporter 1 level in FCSKKO cells, as the entry of fucose was probably not disturbed in this cell line, in contrast to TSTA3KO cells. In comparison to wild-type cells, no changes were observed (Fig 6D). Western blotting of CaSR showed a similar level of protein in TSTA3KO and GMDSKO cells to wild-type cells (Fig 6E), regardless of fucose supplementation. It showed that only the fucose uptake through the GLUT1 transporter was regulated by TSTA3 protein, and macropinocytosis, in which CaSR protein is a receptor that is responsible for initiation of the process, was not affected in either the TSTA3 or GMDSKO cell line.

Having found that fucose uptake may be partially impaired in GMDSKO cells but not in TSTA3KO cells, we wondered whether lysosomal degradation of glycans, which is another source of free fucose, may also be influenced by a lack of particular enzymes from GDP-fucose biosynthesis pathways. In order to do this, we performed Western blotting of FUOM in wild-type, TSTA3KO and GMDSKO cells (Fig 7A), supplemented and not supplemented with fucose. We observed no differences between TSTA3KO and GMDSKO cells, and no dissimilarities compared to wild-type cells. Even if only the salvage biosynthesis pathway could utilize fucose as a substrate, an unchanged level of FUOM protein was observed in FCSKKO cells compared to wild-type cells (Fig 7B). We also evaluated changes in FUCA1 protein level in wild-type, GMDSKO, TSTA3KO and FCSKKO cell lines (Fig 7C, 7D). No changes were observed in FUCA1 production between TSTA3KO, GMDSKO and wild-type cells, regardless of fucose supplementation. Also, the protein level was not changed in the FCSKKO cell line compared to the wild-type cells.

## Discussion

In this study, we examined the impact of the inactivation of particular enzymes from GDP-fucose biosynthesis pathways on fucose metabolism in a model human cell line and their mutual influence on the action of individual pathways. Here, we have presented i) differences in GDP-fucose synthesis between TSTA3KO and GMDSKO cells; ii) possible explanations for abnormal production of GDP-fucose, e.g. the selective influence of deficiency of TSTA3 protein on fucokinase and disturbances in fucose uptake; iii) the mechanism of complementation of GDP-fucose and fucosylated structures' synthesis in cells lacking fucokinase.

We used the human cell line HEK293T for the generation of complete inactivation of genes encoding fucokinase, an enzyme taking part in the salvage biosynthesis pathway of GDP-fucose, and GMDS and TSTA3 proteins, enzymes belonging to the *de novo* biosynthesis pathway of GDP-fucose. It is true that mutations in the tested proteins, TSTA3 and FCSK, do occur. However, when the activity of the mutants was tested, it was reduced but not completely

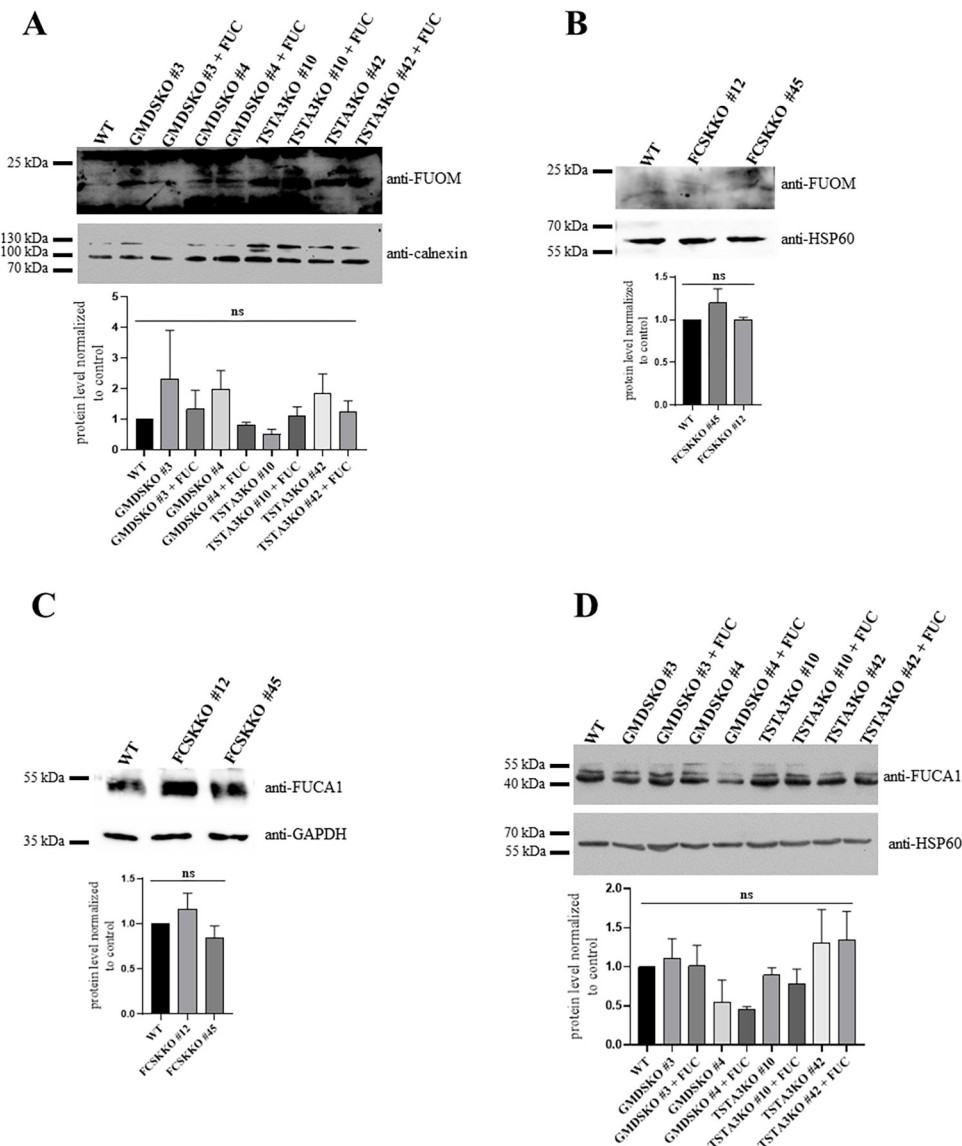

**Fig 7. Influence of TSTA3, GMDS and FCSK protein inactivation on the level of proteins involved in fucose recycling in mammalian cells.** (A) and (B) wild-type, TSTA3KO and GMDSKO HEK293T cells fed and not fed with fucose as well as FCSKKO cells were analysed by western blotting to determine protein levels of FUOM; as a loading control, an anti-calnexin antibody and anti-HSP60 antibody were used. (C) and (D) western blotting of FUCA1 performed on wild-type, TSTA3KO, GMDSKO, supplemented and unsupplemented in fucose and FCSKKO cell lysates. Anti-HSP60 and anti-calnexin antibodies were used as a loading control. For (A) and (D), ns, not significant as determined using one-way ANOVA with the Tukey post-hoc test. For (B) and (C), ns, not significant as determined using Welch's t-test. Data are represented as mean ± SEM. Each sample was run at least in two biological replicates.

absent [32–34]. To the best of our knowledge, the mentioned GDP-fucose level has not been studied before in such a cell model [24–28, 32–37]. Unexpectedly, upon fucose supplementation in the TSTA3KO cell line, abnormal production of GDP-fucose was observed (Fig 2A), and it was specific to the inactivation of this enzyme but not for blocking of the entire *de novo* biosynthesis pathway, because, in the GMDSKO cell line, such induction did not happen (Fig 2C). It is worth noting that a similar pattern of biosynthesis of the nucleotide sugar is exhibited by UDP-galactose production in mammalian cells [38]. The study using knockout of

UDP-galactose 4′ epimerase (GALE) in different human cell lines showed accumulation of nucleotide sugar too, upon galactose supplementation; however, in UDP-galactose biosynthesis, only one enzyme determines the *de novo* pathway. These two effects are very similar, yet they may have completely different causes, because these biosynthetic pathways, for GDP-fucose and UDP-galactose, are distinct and have other mechanisms of conversion of monosugar to nucleotide sugar. On the other hand, differences in the GDP-fucose biosynthesis rate recorded in TSTA3KO and GMDSKO cell lines did not influence the incorporation of fucose into glycoproteins, at least via α-1,6 linkage (Fig 2B and 2D). It seems that only the regulation of nucleotide sugar biosynthesis is disturbed. However, its translocation to the glycosylation site, at least upon 5 mM fucose supplementation, appears unchanged (Fig 2C). Previously, it was reported that the human near-haploid cell line derived from the chronic myelogenous leukaemia cell line (HAP1) deficient in fucokinase was unable to use externally added fucose to N-glycan synthesis [32]. This was demonstrated by the radioactivity measurements of isolated N-glycans after cell supplementation with [5,6-3H]-fucose. However, neither GDP-fucose concentration nor the percentage of fucosylated structures was checked to establish whether this ~10% of the total pool of GDP-fucose disappears or is complemented by the second biosynthetic pathway of GDP-fucose. Analysis of the cellular GDP-fucose concentration in FCSKKO cells showed a slightly lower level, but not statistically significantly, when compared to the cellular concentration in wild-type cells (Fig 2E). It did not affect the amount of fucosylated glycan structures, which was highly similar to that amount in wild-type cells (Fig 2F).

Testing the effects of different fucose concentrations in cell culture media on GDP-fucose level and synthesis of fucosylated structures in TSTA3KO and GMDSKO cell lines provided exciting observations. GDP-fucose synthesis was enhanced among all applied external fucose supplementations of TSTA3KO cells compared to GMDSKO and WT cells. The percentage of fucosylated N-glycans was not reduced in cell lines lacking GMDS protein (Fig 3A and 3B). Even with 5 mM fucose in cell culture media, the percentage was higher than in TSTA3KO cells. Along with the previous observations, GDP-fucose production is affected by the absence of particular enzymes but not its incorporation into glycans. The effect of restoration of fucosylation of glycans in both GMDSKO and TSTA3KO cells in all ranges of applied fucose concentrations could be explained by the presence of SLC35C1 protein, which was not disturbed either in TSTA3KO or GMDSKO cells comparing to wild-type cells (S3A and S3B Fig), and also our previously postulated hypothesis of the existence of at least three routes of delivery of GDP-fucose into the Golgi apparatus [30]. The first one is SLC35C1-dependent and mainly utilises the nucleotide sugar pool derived from the *de novo* pathway. The other two are SLC35C1-independent and mainly use the GDP-fucose pools synthesised by the salvage pathway. However, the first of them works even under very low (nanomolar) concentrations of exogenous fucose, whereas the other requires much higher concentrations of this nucleotide sugar that can only be obtained by feeding the cells with sub-millimolar and millimolar concentrations of fucose. It raises an interesting question: What is the potential reason for the increased nucleotide sugar synthesis in the mentioned cell lines?

The first idea that comes to mind involves changes in the levels of enzymes involved in the salvage biosynthetic pathways. In the case of TSTA3 and GMDS protein inactivation, the level of fucokinase was raised in TSTA3KO cells, especially in those fed with fucose (Fig 4B). A similar study was conducted on the knockout of GALE in different human cell lines [38]. They assessed the levels of galactokinase 1 and 2 (GALK1 and GALK2) in cells deficient in GALE enzyme. No changes were observed in that case. However, overexpression of FPGT and FCSK proteins in GMDSKO and TSTA3KO cells revealed differences in the synthesis of nucleotide sugar and fucosylated structures (Fig 5A and 5B). While the overproduction of FPGT enhanced GDP-fucose and fucosylated structures in both studied cell lines, increased levels of

FCSK raised them only in TSTA3KO cells. In FCSKKO cells, the level of GMDS did not change, but the TSTA3 protein level rose (Fig 4C and 4D). It might come to mind that when TSTA3 is disrupted, the cellular pool of GDP-fucose might signal an upregulation of salvage pathway enzymes, such as fucokinase (FCSK) and GDP-fucose pyrophosphorylase (FPGT), to compensate for the reduced GDP-fucose synthesis.

We also suggest that in mammalian cells, the mechanism of uptake of free monosugars from the environment may also be regulated. For several sugars, i.e. glucose, galactose and fucose, membrane transporters as well as endocytic pathways were described as ways of sugar uptake from outside the cell [22, 39]. It was found that the inactivation of an enzyme constituting part of the *de novo* biosynthesis pathway of UDP-galactose influenced galactose uptake [40]. Our results, presented in Table 1, revealed differences between TSTA3KO and GMDSKO cells in fucose uptake from cell culture media, which was more promoted by the cell line deficient in TSTA3 protein. On the other hand, the lack of fucokinase did not change the uptake when compared to wild-type cells. To the best of our knowledge, this is the first evidence suggesting a connection between enzymes of the *de novo* biosynthesis pathway of nucleotide sugar and sugar imbibition from the extracellular space. Going further, it seems that enhanced fucose uptake depends on the increased level of GLUT1 in TSTA3KO cells compared to GMDSKO cells upon fucose supplementation (Fig 6C), wherein uptake through the mechanism of macropinocytosis was probably not unbalanced in either of the studied cell lines (Fig 6D). Moreover, examination of the level of GLUT1 transporter in cell lines deficient in fucokinase showed no changes compared to the wild-type cells (Fig 6E). Overall, it shows that the *de novo* biosynthesis pathway may regulate the action of the salvage biosynthesis pathway of GDP-fucose by providing substrate to the cells.

Fucose may also be delivered for the salvage pathway of GDP-fucose biosynthesis from the lysosomal degradation of glycans [14], where FUCA1 enzymatically removes α-fucose from glycans, and then FUOM converts it to β-anomer [21]. No changes were noted in the level of either FUCA1 or FUOM in GMDSKO and TSTA3KO cells (Fig 7A and 7C). No differences in levels of FUOM and FUCA1 proteins were observed in FCSKKO cells compared to wild-type cell lines (Fig 7B and 7D). A deficiency of fucokinase, which is the only enzyme in mammalian cells using free fucose as the substrate, did not cause any change in protein levels of enzymes involved in recycling fucosylated glycans and did not influence fucose acquisition from outside the cell. On the other hand, a deficiency of the TSTA3 enzyme impacted the mechanism of delivering fucose from outside the cell, in contrast to the cell line lacking GMDS protein.

Proper localisation of enzymes involved in glycosylation may be crucial for undisturbed glycan assembly [41]. For example, the mislocalisation of Golgi glycosyltransferases caused by modulation of Golgi pH impaired terminal N-glycosylation [42]. Proper localisation could accompany protein interactions [43]. To date, the interactions among nucleotide sugar transporters (NSTs), NSTs and glycosyltransferases have been described [44–48]. Disturbed interactions could cause defects, e.g., sialylation [40]. In terms of interaction between synthases of nucleotide sugars, the formation of complexes between MUR1 and AtTSTA3/GER1 protein, equivalents of GMDS and TSTA3, in *Arabidopsis thaliana* has been experimentally determined [49]. Nakayama et al. studied the activity of MUR1 and AtTSTA3/GER1 overproduced in *Saccharomyces cerevisiae* independently or co-overproduced. They observed no activity of MUR1 overexpressed alone, and it needs to be co-expressed with AtTSTA3/GER1 to gain activity in vivo. Because the created complexes of MUR1 and AtTSTA3/GER1 stabilised the MUR1 protein in cells, its degradation was prevented. However, it concerned the formation of complexes between enzymes in the *de novo* biosynthesis pathway, not the salvage pathway. We assume that by creating complexes with proteins of the salvage biosynthesis pathway, enzymes of the *de novo* biosynthesis pathway could over/downregulate the work of FCSK or FPGT proteins.

GDP-mannose 4,6-dehydratase, the fucokinase and the GDP-keto-6-deoxymannose-3,5-epimerase are enzymes involved in GDP-fucose synthesis in mammalian cells, either salvage or *de novo* pathways [2]. Impairments in genes encoding the latter two proteins lead to congenital disorders of glycosylation (CDG), a family of inherited metabolic diseases that is still rapidly growing [50]. Currently, 6 cases have been diagnosed for defects in the gene encoding fucokinase (FCSK-CDG) [32–35]. Patients usually suffer from severe malformations, e.g. intellectual disability, growth delay, central hypotonia or ophthalmological disorders. Currently, no treatment is available for this condition, so exploring the relationship between the two nucleotide sugar synthesis pathways in depth is essential. The results could lead to the development of a potential treatment. One patient with defective TSTA3 protein has been reported [51]. Another disease connected to fucosylation is caused by mutations in a gene encoding SLC35C1 protein called leukocyte adhesion deficiency type II (LAD II). LAD II leads to a reduction in the fucosylation of glycoconjugate in general, presumably induced by an impaired GDP-fucose transport to Golgi apparatus [3]. Characteristic symptoms of this disease include delayed psychomotor development, the Bombay phenomenon, short stature, immunodeficiency, facial dysmorphism, frequent bacterial infections and leukocytosis. To date, 19 cases of LAD II have been diagnosed [52–60]. The only treatment proposed to patients is fucose supplementation. However, it requires enormous doses of L-fucose and did not help in all cases where oral fucose administration was applied [52, 53, 56–61]. Therefore, it is so important to study the mechanism of GDP-fucose synthesis and the mutual responses between the pathways of GDP-fucose synthesis in mammalian cells. Again, the results obtained from such experiments could improve the current treatment method, helping all patients suffering from LAD II.

Summing up, this study describes relationships between *de novo* and salvage GDP-fucose biosynthesis pathways. Moreover, our results show differences in nucleotide sugar synthesis within the mannose-derived route of nucleotide sugar synthesis and suggest a potential explanation for this phenomenon. GDP-fucose and fucosylated glycan synthesis in the cell line deficient in fucokinase was not disturbed. No aberrations in FPGT level in the TSTA3KO and GMDSKO cell lines were observed. However, an elevated level of FCSK protein was observed in TSTA3KO cells. Additionally, it is notable that overexpression of fucokinase in the TSTA3KO cell line raised GDP-fucose production and fucosylated glycan synthesis, opposite to the result in the GMDSKO cell line. Moreover, the fucose uptake was more pronounced in cells lacking TSTA3KO compared to cells deficient in GMDS. Interestingly, control of fucose metabolism in FCSKKO cells occurred only in the case of a raised level of TSTA3 protein. Overall, the regulation of the salvage pathway in the absence of TSTA3 or GMDS protein appears to be complicated and multi-level and may depend on more than one mechanism.

## Supporting information

**S1 Fig. Confirmation of inactivation of genes encoding proteins involved in GDP-fucose synthesis.** (A) TSTA3 western blotting analysis in wild-type and TSTA3KO HEK293T cell lines. Anti-HSP60B antibody was used as a loading control. (B) GMDS western blotting analysis in wild-type and GMDSKO HEK293T cell lines. An anti-HSP60 antibody was used as a loading control. (C) FCSK western blotting analysis in wild-type and FCSKKO HEK293T cell lines. A green frame indicates an appropriate band coming from FCSK in wild-type cells. The anti-HSP60 antibody was used as a loading control. (D) Verification of a knockout of the TSTA3 gene in HEK293T cell line. Total RNA and genomic DNA (gDNA) were isolated from the wild-type (WT) TSTA3 knockout (TSTA3KO) cells, and either PCR (DNA) or RT-PCR (mRNA) was performed using TSTA3 gene-specific primers. (E) Verification of a knock-out

of the GMDS gene in HEK293T cell line. Total RNA and genomic DNA (gDNA) were isolated from the wild-type (WT) and GMDS knockout (GMDSKO) cells, and either PCR (DNA) or RT-PCR (mRNA) was performed using GMDS gene-specific primers. (F) Verification of a knockout of the TSTA3 gene in HEK293T cell line. Total RNA and genomic DNA (gDNA) were isolated from the wild-type (WT) TSTA3 knockout (TSTA3KO) cells, and either PCR (DNA) or RT-PCR (mRNA) was performed using TSTA3 gene-specific primers. (G) Verification of a knock-out of the FCSK gene in HEK293T cell line. Total RNA and genomic DNA (gDNA) were isolated from the wild-type (WT) and FCSK knockout (GMDSKO) cells, and either PCR (DNA) or RT-PCR (mRNA) was performed using FCSK gene-specific primers.
(TIF)

**S2 Fig. Exemplary chromatograms of HPLC separation of nucleotide sugars.** Cell lysates of (A) HEK293T WT, (B) HEK293T TSTA3KO, (C) HEK293T TSTA3KO with the addition of GDP-fucose standard were subjected to nucleotide sugar extraction, and then RP-HPLC separation. Peaks correspond to GDP-fucose, and unknown ones in TSTA3KO cells are signed and indicated by arrows.
(TIF)

**S3 Fig. Estimation of SLC35C1 protein level in HEK293T wild-type and deficient in TSTA3 and GMDS proteins cell lines.** Cell lysates were applied for western blotting of SLC35C1 protein in (A) TSTA3KO and (B) GMDSKO cell lines either fed with fucose or not and compared to wild-type cells. Ponceau S staining was used as a loading control for all western blotting experiments. Data are represented as mean ± SEM. Each sample was run at least in three biological replicates. ns, not significant, as determined using one-way ANOVA with the Tukey post-hoc test.
(TIF)

**S1 File. Compressed file with crude data.**
(ZIP)

**S2 File. List of primers and antibodies used in this study.**
(DOCX)

## Author Contributions

**Conceptualization:** Edyta Skurska, Mariusz Olczak.

**Data curation:** Edyta Skurska.

**Formal analysis:** Edyta Skurska, Mariusz Olczak.

**Funding acquisition:** Edyta Skurska, Mariusz Olczak.

**Investigation:** Edyta Skurska, Mariusz Olczak.

**Methodology:** Edyta Skurska.

**Project administration:** Mariusz Olczak.

**Supervision:** Mariusz Olczak.

**Writing – original draft:** Edyta Skurska.

**Writing – review & editing:** Mariusz Olczak.

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
