## [Decision Letter · Decision Letter 0]

17 Jul 2024

PONE-D-24-22138Interplay between de novo and salvage pathways of GDP-fucose synthesisPLOS ONE

Dear Dr. Olczak,

Thank you for submitting your manuscript to PLOS ONE. After careful consideration, we feel that it has merit but does not fully meet PLOS ONE’s publication criteria as it currently stands. Therefore, we invite you to submit a revised version of the manuscript that addresses the points raised during the review process.

We look forward to receiving your revised manuscript.

Kind regards,

Ashutosh Pandey, Ph.D.

Academic Editor

PLOS ONE

Journal Requirements:

This research was funded by the National Science Centre (Narodowe Centrum Nauki, NCN), Poland, grant number 2022/45/N/NZ3/00093

3. Please upload a new copy of Figures 1-7 and S1_Figure as the detail is not clear. Please follow the link for more information: " ext-link-type="uri" xlink:type="simple">https://blogs.plos.org/plos/2019/06/looking-good-tips-for-creating-your-plos-figures-graphics/"
" ext-link-type="uri" xlink:type="simple">https://blogs.plos.org/plos/2019/06/looking-good-tips-for-creating-your-plos-figures-graphics/"

4. We notice that your supplementary tables are included in the manuscript file. Please remove them and upload them with the file type 'Supporting Information'. Please ensure that each Supporting Information file has a legend listed in the manuscript after the references list.

Reviewers' comments:

Reviewer's Responses to Questions

**Comments to the Author**

1. Is the manuscript technically sound, and do the data support the conclusions?

Reviewer #1: Yes

Reviewer #2: Yes

2. Has the statistical analysis been performed appropriately and rigorously? 

Reviewer #1: Yes

Reviewer #2: Yes

3. Have the authors made all data underlying the findings in their manuscript fully available?

Reviewer #1: Yes

Reviewer #2: Yes

4. Is the manuscript presented in an intelligible fashion and written in standard English?

Reviewer #1: Yes

Reviewer #2: Yes

5. Review Comments to the Author

**Reviewer #1:** The work reported by Edyta Skurska et al., titled "Interplay between de novo and salvage pathways of GDP-fucose synthesis," offers valuable insights into how enzymes are regulated in human knockout cell lines and their significance in fucose metabolism. It provides a comprehensive exploration of the interaction between different pathways involved in GDP-fucose synthesis, shedding light on their roles in cellular processes and potential implications for health and disease.

The authors investigated the interdependency of various enzymes within the two primary pathways of GDP-fucose synthesis. Their study adeptly utilized in vivo assays on human HEK293T cell lines, where they generated knockouts of key enzymes from both the de novo and salvage pathways.

This manuscript primary assertion is that previous studies overlooked the interaction between the de novo and salvage pathways of GDP-fucose synthesis. This study represents the first comprehensive report elucidating the roles of enzymes crucial to GDP-fucose production. The authors successfully demonstrated GDP-fucose formation and observed the response of fucosylated structures in knockout strains when supplemented with free fucose.

Overall, the study illuminates the roles of three key enzymes—GMDS, TSTA, and fucokinase—in knockout strains. Despite blocking the expression of the major salvage pathway enzyme, fucokinase, the levels of GDP-fucose and fucosylated glycans remained unchanged. The experimental findings are robustly supported by the presented data and are effectively communicated in the article.

Minor points :

• In Page 8, line no. 172, The author states that the GDP-fucose concentration in unfed TSTA3KO cells was estimated as 0 µM. What is the inherent cellular concentration of GDP fucose without external supplementation?

• Observed the presence of multiple bands in FCSK knockout cell line (figure 4D)

• "What methods are available to quantify the reaction intermediate GDP-4-keto-6-deoxy mannose in biochemical studies?"

• Is it possible to visually detect the upregulation or downregulation of SLC35C1 in knockout cell lines

**Reviewer #2:** The manuscript highlights fucose as a critical component of glycoconjugates involved in numerous biological processes in mammalian cells, such as cell adhesion, tissue development, angiogenesis, fertilization, malignancy, and tumor metastasis. The active form of fucose, GDP-fucose, is synthesized through two main pathways: the de novo pathway and the salvage pathway. They describe the enzymes involved in both pathways. The study used the CRISPR/Cas9 system to create cell lines deficient in TSTA3 (TSTA3KO), GMDS (GMDSKO), and FCSK (FCSKKO) in HEK293T cells. Their key finding and novelty is that they highlight a mutual regulation between de novo and salvage pathway enzymes. Results that lead to this finding are:

• Cells lacking TSTA3 produced high amounts of GDP-fucose upon fucose supplementation, unlike GMDS-deficient cells.

• After fucose supplementation, fucokinase levels were elevated in TSTA3-deficient cells, but not in GMDS-deficient cells.

• TSTA3 protein levels increased in FCSK-deficient cells.

• Differences in fucose uptake were observed between the studied cell lines.

The manuscript is acceptable after a few minor changes.

Comments:

Introduction section: The manuscript will significantly benefit by providing more context about the significance of fucose in glycoconjugates and why its study is essential. Consider elaborating on how defects in fucosylation can lead to diseases, which would emphasize the biological importance and relevance of the study.

Figure legend Fig 1B: Line 35: 'graph illustrating the fucose-derived pathway of GDP-fucose synthesis.' Fig 1B is also a linear schematic representation of the Salvage pathway. Why do the authors call it a graph?

Line 45: 'Recently, the second mechanism of fucose delivery, depending on glucose transporter 1 (GLUT1) activity, was described'. I think the authors should use the word 'uptake' instead of delivery to maintain uniformity.

Line 47: 'The depletion of genes encoding enzymes of the de novo GDP-fucose biosynthesis pathway was applied to produce afucosylated antibodies.' This line seems to disturb the flow of the manuscript. Consider beginning the sentence like 'To illustrate the importance of fucosylated antibodies…'

Material and Methods section:

Line 93: 'According to the manufacturer's protocol, HEK293T GMDSKO and FXKO cell lines.' Do the authors mean the FCSKKO cell line? If so, please use FCSKKO throughout the manuscript. Ensure to check Table 1 as well.

Results section:

Line 191: 'GMDS is the second enzyme of the de novo biosynthesis pathway.' GMDS is the first enzyme.

Line 215: 'The addition of fucose in the concentration of 10 μM already caused an increase in GDP-fucose levels in TSTA3KO cells to the rate between ~100 and 600 μM.' Is this statement in the context of Figure 3A? If so, it is unclear which histogram is being talked about. I think you are referring to the third histogram in the graph, which corresponds to 50 uM fucose. Is that right?

The authors mention a number besides the TSTA3KO and GMDSKO lines throughout the figures. It would be helpful to mention the context of these numbers once at the beginning of the results section. Are these different cell lines or just a different replica?

Line 247: 'However, the FCSK protein level was elevated in one of the clones of TSTA3KO cells, regardless of fucose supplementation'. For Fig 4: Please mark an arrow or box in the western blots on the lane you want the readers' attention to.

Why do you observe two bands for FPGT (Fig 4A)?

Figure 5: The figure shows 'HAFUK' and 'cmycFPGT'. Please briefly mention these tags/constructs in the results so it will be easier to follow the western blots.

Figure 7 A and B: Do you have better-looking western blots?

Discussion section:

Feedback Regulation Mechanism:

TSTA3 might be involved in feedback regulation where the levels of GDP-fucose produced via the de novo pathway influence the activity or expression of enzymes in the salvage pathway.

When TSTA3 activity is disrupted, the cellular pool of GDP-fucose might signal an upregulation of salvage pathway enzymes, such as fucokinase (FCSK) and GDP-fucose pyrophosphorylase (FPGT), to compensate for the reduced GDP-fucose synthesis.

If the authors could comment on the above in their discussion, it would enhance the manuscript's readability.

Does TSTA3 form complexes with other enzymes involved in GDP-fucose biosynthesis, stabilizing them and enhancing their activities?

Does TSTA3 influence the transcription of genes encoding enzymes of both pathways?

6. PLOS authors have the option to publish the peer review history of their article (what does this mean?). If published, this will include your full peer review and any attached files.

Reviewer #1: No

Reviewer #2: No

---

## [Author Response · Author response to Decision Letter 0]

2 Aug 2024

Answers to editor points and reviewers’ comments

A: Thank you very much for the critical review of our manuscript. We agree with the majority (but not all) of the received comments. We tried to overcome those problems with a newly performed experiment and by improving the text of the manuscript to be available for a broader audience. Also, supplementary material was changed and completed with new figures, according to the reviewer’s suggestions. Below, please find our detailed explanations for each comment expressed in the first version of our manuscript.

A. Answers to editor points

Journal Requirements:

A: The style was checked and corrected according to templates, available at the journal website.

This research was funded by the National Science Centre (Narodowe Centrum Nauki, NCN), Poland, grant number 2022/45/N/NZ3/00093

A: We would like to add an additional source of funding. Now, two grants should be listed, could you please include this information in the final version of our submission:

National Science Centre (Narodowe Centrum Nauki, NCN), Poland, grant number 2022/45/N/NZ3/00093 (to ES) (this is already present in the uploading system)

and

National Science Centre (Narodowe Centrum Nauki, NCN), Poland, grant number 2023/51/B/NZ3/00810 (to MO) (the new source of financing)

In the cover letter, we declared that the funders had no role in study design, data collection and analysis, decision to publish, or preparation of the manuscript in both grants. 

3. Please upload a new copy of Figures 1-7 and S1_Figure as the detail is not clear. Please follow the link for more information: https://blogs.plos.org/plos/2019/06/looking-good-tips-for-creating-your-plos-figures-graphics/" https://blogs.plos.org/plos/2019/06/looking-good-tips-for-creating-your-plos-figures-graphics/"

A: All figures, including new ones (in Supplementary material), were corrected. according to PLOS One website suggestions.

4. We notice that your supplementary tables are included in the manuscript file. Please remove them and upload them with the file type 'Supporting Information'. Please ensure that each Supporting Information file has a legend listed in the manuscript after the references list.

A: We transferred tables to Supplementary material. Also, legends for supporting information files are now present at the end of the manuscript.

A: Now, the original uncropped and unadjusted images to all of electrophoresis and blotting experiments reported in the submission’s figures are uploaded. All files (with the exception of supplementary figures, which were uploaded separately) were compressed in the separate folder which also contains numerical, raw data (Excel file), and a big set of chromatograms, converted to in ASCII format, which can be easily read with programs like Excel, Origin, Statistica, and many others.

A: Checked and corrected. Because the text of our manuscript was significantly changed, the order in reference list was also changed.

B. Answers to Reviewers' comments:

Reviewer #1

A: Thank you very much for the critical review of our manuscript. We agree with the majority (but not all) of the received comments. We tried to overcome those problems with a newly performed experiment and also by correcting the text of the manuscript to be available for a broader audience. Also, supplementary material was improved and completed with new figures, according to the reviewer’s suggestions

The work reported by Edyta Skurska et al., titled "Interplay between de novo and salvage pathways of GDP-fucose synthesis," offers valuable insights into how enzymes are regulated in human knockout cell lines and their significance in fucose metabolism. It provides a comprehensive exploration of the interaction between different pathways involved in GDP-fucose synthesis, shedding light on their roles in cellular processes and potential implications for health and disease.

The authors investigated the interdependency of various enzymes within the two primary pathways of GDP-fucose synthesis. Their study adeptly utilized in vivo assays on human HEK293T cell lines, where they generated knockouts of key enzymes from both the de novo and salvage pathways.

This manuscript primary assertion is that previous studies overlooked the interaction between the de novo and salvage pathways of GDP-fucose synthesis. This study represents the first comprehensive report elucidating the roles of enzymes crucial to GDP-fucose production. The authors successfully demonstrated GDP-fucose formation and observed the response of fucosylated structures in knockout strains when supplemented with free fucose.

Overall, the study illuminates the roles of three key enzymes—GMDS, TSTA, and fucokinase—in knockout strains. Despite blocking the expression of the major salvage pathway enzyme, fucokinase, the levels of GDP-fucose and fucosylated glycans remained unchanged. The experimental findings are robustly supported by the presented data and are effectively communicated in the article.

Minor points :

• In Page 8, line no. 172, The author states that the GDP-fucose concentration in unfed TSTA3KO cells was estimated as 0 µM. What is the inherent cellular concentration of GDP fucose without external supplementation?

A: The concentrations of GDP-fucose were calculated from chromatograms after ion-pairing chromatography separations of nucleotides previously extracted and purified from cell lysates. The GDP-fucose was identified as a peak with a retention time identical to the GDP-fucose standard (Rt=30.4 minutes). However, in the case of TSTA3KO, we observed only one peak, which was adjacent closely (Rt=30.2 min) to the (potentially) expected GDP-fucose (the last one was not visible). To distinguish between these two peaks, in one of our separations, we added external GDP-fucose during chromatography of nucleotides extracted from TSTA3KO cells and observed two distinct peaks. Probably, the unique, new peak of Rt=30.2 minutes, belongs to the accumulated, intermediate product of GMDS action, the 1st enzyme of the de novo pathway (GDP-4-keto-6-deoxymannose). However, we did not confirm that, but it is clear, that this peak was not GDP-fucose. Unfortunately, the presence of the additional peak makes the estimation of GDP-fucose concentration not easy. The peak of the retention time identical to standard GDP-fucose was not visible. Therefore, we assumed that it was not present. Unfortunately, it cannot be ruled out, that very small amounts of GDP-fucose were masked by the adjacent, unknown peak, present after the separation of nucleotides from TSTA3KO cells only.

In the revised version of our manuscript, for clarity, in supplementary material, we added chromatograms of TSTA3KO nucleotide pool, with the explanation of this phenomenon (see new Figure S2).

• Observed the presence of multiple bands in FCSK knockout cell line (figure 4D)

A: The antibodies were not absolutely specific. Unspecific bands are also present, but now the real TSTA3 band, which is very changed in FCSKKO cell line is additionally marked with the frame

• "What methods are available to quantify the reaction intermediate GDP-4-keto-6-deoxy mannose in biochemical studies?"

A: It may be done after chromatography separation. However, it would not be easy. There is no GDP-4-keto-6-deoxy mannose available on the market (usually used as a standard for quantification). Additionally, it seems that this compound is not very stable in solutions. There are also colorimetric methods, however, not very sensitive. In the near future, we plan to introduce a new method for enzymatic, in vitro tests of purified, recombined GMDS activity, probably with the use of radioactive GDP-mannose.

• Is it possible to visually detect the upregulation or downregulation of SLC35C1 in knockout cell lines

A: We agree that this problem is very interesting. Because of that, we performed a new experiment and checked the level of SLC35C1 in the GMDSKO and TSTA3KO cell lines. We did not recognize statistically important changes. The result of this experiment is present in Supplementary Figure 3.

Reviewer #2: 

A: Thank you very much for the critical review of our manuscript. We agree with the majority (but not all) received comments. We tried to overcome those problems with a newly performed experiment and also by correcting the text of the manuscript to be available for a broader audience. Also, supplementary material was improved and completed with new figures, according to the reviewer’s suggestions

The manuscript highlights fucose as a critical component of glycoconjugates involved in numerous biological processes in mammalian cells, such as cell adhesion, tissue development, angiogenesis, fertilization, malignancy, and tumor metastasis. The active form of fucose, GDP-fucose, is synthesized through two main pathways: the de novo pathway and the salvage pathway. They describe the enzymes involved in both pathways. The study used the CRISPR/Cas9 system to create cell lines deficient in TSTA3 (TSTA3KO), GMDS (GMDSKO), and FCSK (FCSKKO) in HEK293T cells. Their key finding and novelty is that they highlight a mutual regulation between de novo and salvage pathway enzymes. Results that lead to this finding are:

• Cells lacking TSTA3 produced high amounts of GDP-fucose upon fucose supplementation, unlike GMDS-deficient cells.

• After fucose supplementation, fucokinase levels were elevated in TSTA3-deficient cells, but not in GMDS-deficient cells.

• TSTA3 protein levels increased in FCSK-deficient cells.

• Differences in fucose uptake were observed between the studied cell lines.

The manuscript is acceptable after a few minor changes.

Comments:

Introduction section: The manuscript will significantly benefit by providing more context about the significance of fucose in glycoconjugates and why its study is essential. Consider elaborating on how defects in fucosylation can lead to diseases, which would emphasize the biological importance and relevance of the study.

A: We added paragraphs to the text in Introduction and Discussion sections.

Figure legend Fig 1B: Line 35: 'graph illustrating the fucose-derived pathway of GDP-fucose synthesis.' Fig 1B is also a linear schematic representation of the Salvage pathway. Why do the authors call it a graph?

A: It was corrected. Now, this is called ‘diagram’.

Line 45: 'Recently, the second mechanism of fucose delivery, depending on glucose transporter 1 (GLUT1) activity, was described'. I think the authors should use the word 'uptake' instead of delivery to maintain uniformity.

A: Corrected.

Line 47: 'The depletion of genes encoding enzymes of the de novo GDP-fucose biosynthesis pathway was applied to produce afucosylated antibodies.' This line seems to disturb the flow of the manuscript. Consider beginning the sentence like 'To illustrate the importance of fucosylated antibodies'

A: Corrected.

Material and Methods section:

Line 93: 'According to the manufacturer's protocol, HEK293T GMDSKO and FXKO cell lines.' Do the authors mean the FCSKKO cell line? If so, please use FCSKKO throughout the manuscript. Ensure to check Table 1 as well.

A: Corrected in the text and in Table 1, as well.

Results section:

Line 191: 'GMDS is the second enzyme of the de novo biosynthesis pathway.' GMDS is the first enzyme.

Line 215: 'The addition of fucose in the concentration of 10 μM already caused an increase in GDP-fucose levels in TSTA3KO cells to the rate between ~100 and 600 μM.' Is this statement in the context of Figure 3A? If so, it is unclear which histogram is being talked about. I think you are referring to the third histogram in the graph, which corresponds to 50 uM fucose. Is that right?

A: It was checked and now is more precisely pointed

The authors mention a number besides the TSTA3KO and GMDSKO lines throughout the figures. It would be helpful to mention the context of these numbers once at the beginning of the results section. Are these different cell lines or just a different replica?

A: Corrected. Of course, these are not replica. Now, separate clones of the same KO experiment are numbered

Line 247: 'However, the FCSK protein level was elevated in one of the clones of TSTA3KO cells, regardless of fucose supplementation'. For Fig 4: Please mark an arrow or box in the western blots on the lane you want the readers' attention to.

A: The right band of the expected molecular weight are now marked by frame

Why do you observe two bands for FPGT (Fig 4A)?

A: The band which belongs to FPGT is now marked. Antibodies used in the experiment were not absolutely specific; other band(s) were sometimes visible.

Figure 5: The figure shows 'HAFUK' and 'cmycFPGT'. Please briefly mention these tags/constructs in the results so it will be easier to follow the western blots.

A: Both proteins were produced as fusion molecules, with HA or c-myc peptides, attached to N-termini. We modified the text, as suggested.

Figure 7 A and B: Do you have better-looking western blots?

A: No, it is not easy to find high-quality antibodies on the market. We tried many of them, including mono- and polyclonal antibodies from leading vendors, like Abcam, Santa Cruz or Proteintech, but all of them failed in detections of examined protein (FUOM). The only acceptable Ab was purchased from Cusabio. However, we agree that the quality was far from perfect.

Discussion section:

Feedback Regulation Mechanism:

TSTA3 might be involved in feedback regulation where the levels of GDP-fucose produced via the de novo pathway influence the activ

---

## [Editor Report · Decision Letter 1]

13 Aug 2024

Interplay between de novo and salvage pathways of GDP-fucose synthesis

PONE-D-24-22138R1

Dear Dr. Olczak,

We’re pleased to inform you that your manuscript has been judged scientifically suitable for publication and will be formally accepted for publication once it meets all outstanding technical requirements.

Kind regards,

Ashutosh Pandey, Ph.D.

Academic Editor

PLOS ONE
---

## [Editor Report · Acceptance letter]

19 Aug 2024

PONE-D-24-22138R1 

PLOS ONE

Dear Dr. Olczak, 

I'm pleased to inform you that your manuscript has been deemed suitable for publication in PLOS ONE. Congratulations! Your manuscript is now being handed over to our production team.

Kind regards, 

on behalf of

Dr. Ashutosh Pandey 

Academic Editor

PLOS ONE